# Beehive-Inspired Information Gathering with a Swarm of Autonomous Drones

**DOI:** 10.3390/s19194349

**Published:** 2019-10-08

**Authors:** Alberto Viseras, Thomas Wiedemann, Christoph Manss, Valentina Karolj, Dmitriy Shutin, Juan Marchal

**Affiliations:** German Aerospace Center (DLR), Oberpfaffenhofen, 82234 Weßling, Germany; Thomas.Wiedemann@dlr.de (T.W.); Christoph.Manss@dlr.de (C.M.); valentina.karolj@dlr.de (V.K.); Dmitriy.Shutin@dlr.de (D.S.); Juan.MarchalGomez@dlr.de (J.M.)

**Keywords:** information gathering, multi-robot systems, autonomous robots, drones

## Abstract

This paper presents a beehive-inspired multi-agent drone system for autonomous information collection to support the needs of first responders and emergency teams. The proposed system is designed to be simple, cost-efficient, yet robust and scalable at the same time. It includes several unmanned aerial vehicles (UAVs) that can be tasked with data collection, and a single control station that acts as a data accumulation and visualization unit. The system also provides a local communication access point for the UAVs to exchange information and coordinate the data collection routes. By avoiding peer-to-peer communication and using proactive collision avoidance and path-planning, the payload weight and per-drone costs can be significantly reduced; the whole concept can be implemented using inexpensive off-the-shelf components. Moreover, the proposed concept can be used with different sensors and types of UAVs. As such, it is suited for local-area operations, but also for large-scale information-gathering scenarios. The paper outlines the details of the system hardware and software design, and discusses experimental results for collecting image information with a set of 4 multirotor UAVs at a small experimental area. The obtained results validate the concept and demonstrate robustness and scalability of the system.

## 1. Introduction

In response to natural or technological disasters, UAVs are becoming an increasingly valuable asset for experts and first-responder teams [1]. UAVs provide them with up-to-date surveillance data, enable communication, act as sensor-carrier platforms, and support decision-making in different respects. The use of UAVs, and in particular of a swarm of UAVs, to address the needs of emergency responders, is the main focus of this work.

In general, there are two classes of UAVs that can be deployed for emergency response. The first type are fixed-wing UAVs that use wings to generate a lift. Fixed-wing UAVs are well suited for wide-area operational environments: they can stay airborne over prolonged periods of time ranging in hours. This permits extending the response operational area from hundreds to thousands of square kilometers, depending on the particular UAV model. Such large distances are characteristic for large geological, manmade, or environmental incidents, e.g., earthquakes, large forest fires, flooding, etc. Even relatively small fixed-wing UAVs demonstrate a significant endurance in this respect. However, due to the dynamical constraints of such systems—a fixed-wing UAV typically has to execute a specific flight trajectory—they are typically deployed at heights that are obstacle free, which is typically in the range from 50 m to 100 m, depending on the local regulations.

The second type are rotary-wing UAVs, which have multiple rotors, can fly and maneuver at low altitudes, and can hover near structures. These features make such UAVs appropriate for so-called local-area operational environments [1,2]. Typically, rotary-wing UAVs missions include structural inspections, damage assessments, reconnaissance missions, and mapping applications. In addition, rotary-wing UAVs operate at heights ranging from 3 m to 45 m to give a more detailed coverage of the area of interest, and they travel several kilometers in distance [1]. Thus, rotary-wing UAVs can be used to provide a responder team with a “close-up shot” of the incident scenario. Clearly, a proximity to collapsed structures or debris poses a risk of both the UAV, as well as for the infrastructure or people on the scene, which often requires a dedicated safety officer during operations [3]. As it has been pointed out in [1], the typical human-to-robot-ratio is approx. 3:1, i.e., per one UAV, a single pilot, a safety officer, and a mission specialist (responsible for collecting the data and advising the pilot) are required. To increase the efficiency, usability, and acceptance of UAVs for emergency response, a system is required that on the one hand, provides increased robustness and efficiency with respect to mission goals, while on the other hand, reduces the human-to-robot ratio to a minimum.

A solution to the aforementioned challenges advocated in this work is based on using an autonomous swarm of UAVs, which we understand as a multi-agent system. A swarm consisting of multiple agents can offer a higher efficiency, as tasks can now be shared among members of the swarm. In particular, multiple agents can complete inspection tasks faster compared to a single agent. Also, a higher robustness of the whole system can be achieved since the failure of individual units does not lead to the total system collapse. Moreover, the cost of an individual drone in a swarm can be reduced, as one can trade-off (within certain limits) sensor quality/price and the number of sensing platforms (sensing aperture). Consequently, a potential accidental loss of a drone will come at a lower price. Furthermore, a high level of autonomy makes the system less dependent of the supervision of a human operator. Hence, the human-to-robot ratio can be significantly reduced.

The use of a swarm of robots for monitoring and data collection tasks is not entirely new, though. Previous works also tackle the coverage problem considered in this paper, i.e., reaching certain point of interest (POIs), and taking some action at those points, such as take an image or take a measurement. The coordination of swarms can be solved using biologically inspired approaches [4,5,6,7]. Alternatively, other criteria for path-planning and coordination can be used, as in e.g., [8], where the state-of-health of Lithium Polymer (LiPo) battery is proposed, or more classical task allocation problems for multiple agents using distributed constraint optimization [9] or bounty-based methods [10], to name a few. However, most classical multi-agent approaches for monitoring and data collection imply that individual agents are able to coordinate their actions or even cooperatively solve some underlying optimization problem via communication links. In the literature, this problem is typically addressed in the context of Flying Ad Hoc Network (FANETs) [11]. FANET algorithms focus on how to exchange data within a network composed by multiple drones via drone-to-drone communication links. From a practical perspective an availability of such communication links might pose a problem. Indeed, cost-efficient WiFi links lack coverage, which reduces maximum separation between neighboring agents. Mobile data networks, such as 2G/3G/4G, might introduce significant delays; also, network coverage can be insufficient in some remote operation areas. Dedicated communication links (See, for instance, solutions offered by Mobilicom: https://www.mobilicom.com/) do present a viable solution, yet are rather costly, and require drones with high payload capacity. The latter additionally increases the per-drone price tag of the resulting system.

Alternatively, the coordination can be solved using pre-planned path-planning strategies (see [12] for a good overview). This often results in splitting of an area into several sub-regions or cells, which do not intersect and are obstacle free. Given such split structure of the area of interest, drones’ routes can then be optimally pre-planned to avoid collisions. This alleviates the need for active collision avoidance or peer-to-peer communications, and, consequently, simplifies the whole system design. Likewise, in this paper, we apply the path-planning before flying.

Therefore, the contribution of this paper is (i) to design a swarm-based concept for surveillance and information gathering and (ii) to study the performance of the resulting system in a field experiment. Our intention is to design a simple, cost-efficient, yet robust swarm system that can perform data collection autonomously in (approximately) time-invariant tasks. The proposed swarm concept inherits some of its features from nature, in particular from swarms of bees. In our setting individual UAVs mimic the roles of bees: they collect sensor data—the “nectar”—and bring it to a computational center—the “beehive”. The computational center implements the central data storage and the UAV coordination center, from which the operation of the swarm is controlled and monitored. The flight, data collection, and drone coordination is done in autonomous fashion such that almost no human interaction is required.

The key advantage of our system is the fact that the minimum requirements of a drone’s hardware can be significantly reduced. In particular, the “beehive” can be implemented with a simple Raspberry Pi (https://www.raspberrypi.org/) computer and off-the-shelf WiFi access points, through which the drones exchange data and receive tasks. Similarly, the individual drones do not require an expensive computer or a long-range communication device. They only need the capability to download flight routes and upload the collected data to the “beehive”. The drones merely need to enter the range of the “hive’s” WiFi access point. Additionally, our system does not require an active collision avoidance mechanism consisting of perception sensors and corresponding data processing units. This further simplifies the design of the whole system, and allows for lighter, smaller, and cheaper drones. Of course, the price of this solution is that drones can only communicate in the vicinity of the central station. As a consequence, scenarios where an immediate response to an event is needed the absence of the direct communication is a disadvantage and other solutions might be considered. Yet in no-so-time-critical cases, such as inspection, monitoring, or package delivery, the proposed solution remains a sensible trade-off. To address absence of direct drone-to-drone communication we developed a swarm proactive collision avoidance mechanism. It is important to remark that this mechanism only avoids collisions provided the following assumptions: (i) drones localization accuracy is sufficient to guarantee no inter-drones collisions, (ii) the map of the region of interest ROI contains all static obstacles present in the area, and (iii) there exist no dynamic obstacles in the ROI. This assumption actually holds for many applications of interest. Of course, we could enhance our system by including situational awareness mechanisms by means of additional sensors, such as e.g., light detection and ranging (LIDAR), stereo cameras, etc., together with additional reactive collision avoidance algorithms. This is however out of the scope of this work.

We tested our system in field experiments with 4 UAVs and a single human operator to explore an area of approx. 200×200 m. The collected results show high robustness and efficiency of the whole system. Please note that although the development and experimental validation of the whole system is done with a focus on UAVs, extensions to fixed-wing systems are quite straightforward. Moreover, although the proposed concept avoids using long-range drone-to-drone links, in some time-critical scenarios, such as search and rescue, a long-range low data-rate sensor (e.g., LoRa (https://lora-alliance.org/)) can be used on drones to send a geo-referenced alarm signal to the “beehive”; this LoRa signal then plays the role of a “nectar” the rescue team is interested in. Such an alarm signal can trigger other, more appropriate systems, such as automated package delivery drone to the recipient or dispatch of a rescue team to the alarm location. The proposed concept, however, remains largely unmodified.

The rest of the paper is organized as follows. In Section 2, we provide an overview of our system design as well as a summary of the system’s workflow. This is followed in Section 3 and Section 4 by a detailed description of the key modules that constitute the base station and drones’ system, respectively. Then we provide in Section 6 details about the system implementation from both software and hardware perspective. To finalize, Section 7 describes and analyzes the results of several outdoor experiments that we carried out to evaluate the system proposed in this paper.

## 2. System Overview and Workflow

The proposed system is designed to support first responders during emergency response situations by gathering information with multiple autonomous drones. We take inspiration in nature to design a system that is composed of two main components: a base station that serves as a central “beehive”, and a set of drones that play the role of “bees”. In Figure 1, we depict a block diagram that summarizes the main elements of our system. In the following, we give a general overview of the key system sub-components and implemented workflow, while providing references to individual sections with more detailed descriptions.

**Base station.** The base station consists of three elements: a laptop/tablet computer (Section 3.1), a database (DB) (Section 3.3), and a communication system (Section 3.2).

The computer serves as an interface with a human operator through a graphical user interface (GUI), which we term “Drones Monitoring and Data Visualization” module (Section 3.1.3). In particular, the GUI permits a human operator to select a ROI where to gather information. In addition, it displays the status of the drones and their gathered information, e.g., pictures captured at specific POIs. The computer also runs two additional modules that constitute the “brain” of the base station. These are the “Map Discretization” (Section 3.1.1) and “Routes Computation” (Section 3.1.2) modules. They take a ROI defined by an operator to calculate collision-free flying routes that cover the whole ROI. To this end the ROI is split into smaller regions. Because each of the smaller regions is only assigned to a single drone and drones always stay in their own region, they do not require a reactive collision avoidance. This allows the drones to fly autonomously without relying on a constant communication with the base station or with the other drones. In fact, our system only relies on communication in the vicinity of the “beehive”—base station.

The base station also stores a DB. The DB has two crucial functionalities. First, it stores the information gathered by drones, e.g., pictures or sensor measurements taken at specific POIs. Second, it coordinates the assignment of drones’ flying routes. This avoids that two drones select an identical flying route, which could lead to a collision during flight.

The third element of the base station is a communication system. The communication system works as an interface between the DB and drones. The system design does not require a permanent connection between the base station and the drones, which allows for an off-the-shelf WiFi access point as communication system, instead of costly long-range communication links.

**Drones.** In the used setting, drones play the roles of bees: they collect sensor data—the “nectar”—and bring it back to the central “beehive”. In this respect, our system is specifically designed to be flexible such as to allow using any drone on the market that is equipped with the following components: (i) an onboard computer that incorporates an autopilot functionality to fly to a POI without the need for a human pilot, together with a global positioning system (GPS) module to accurately localize the drone (Section 4.3), (ii) a communication system to transfer information between drones and the DB (Section 4.2), and (iii) an appropriate sensor stack (Section 4.1). Depending on the application, different sensors can be used, such as gas concentration sensors for environmental monitoring, hyperspectral cameras for smart agriculture, visual cameras for inspection and surveillance, or LIDAR sensors for terrain mapping, to name only a few.

The base station and drones constitute the two main elements of our system. Next, we describe the system workflow for both, base station and drones, which includes the following steps.


**System workflow—Base station**


An operator opens the system’s GUI, which automatically displays a world map. Given the map, the operator introduces: the coordinates that specify the ROI in which to gather information, and the drones starting position.With the information about the ROI provided, the "Map Discretization" module loads a digital elevation map (DEM) of the region to identify the areas within the ROI that do not incur a collision with obstacles. Please note that DEM of a region can be downloaded from Google or Open Street Maps, or can be obtained from satellite radar measurements [13] or aircraft LIDAR measurements [14].Next, after the DEM is provided, the "Map Discretization" module executes a map grid-less discretization algorithm that optimally calculates POIs. The POIs are clustered into regions to guarantee a collision-free flight. At this stage, the operator can tune some basic parameters to modify the generation of POIs, e.g., to modify the separation between neighboring POIs, obstacles and other agents.The set of POIs, calculated by the "Map Discretization" module, is the input to our "Routes Computation" module. This takes the POIs and calculates flying routes for the drones that achieve an efficient area coverage.Finally, the flying routes are stored in the DB. At later stages, the routes are accessed by drones through the communication system.

Once the base station workflow (see Figure 2) is terminated, the operator proceeds to activating the drones. This is the only operation that needs to be performed by the operator to control the drones. Once active, each of the drones automatically executes the following workflow:


**System workflow—Drones**


A drone registers itself in the base station, and, then, requests a route from the DB.An unassigned route will be assigned to the drone. If there is no route available, the drone declares itself as "spare" and waits until a route becomes available.For drones that were assigned to a route, the following actions are performed: (i) take-off; (ii) follow the assigned route, while gathering information until a time threshold is exceeded (calculated according to the drone’s battery life); (iii) return to “hive” following a shortest path route calculated from the drone’s assigned POIs.Upon return to “hive”, the drone lands at its take-off position and uploads the information gathered during flight to the DB.Once these steps are finalized the drone automatically disconnects from the system. At this point the operator has the possibility to replace the drone’s battery and activate the drone once more. This will re-trigger the drone’s workflow.

## 3. System Components—Base Station

In this section, we introduce the key modules that constitute the base station. First, we summarize in Section 3.1 the modules that are part of the computer. This is followed by a description of the communication system and the DB in Section 3.2 and Section 3.3, respectively. We summarize in Figure 2 the base station workflow.

### 3.1. Portable Computer

The main modules of the portable computer are the “Grid-Less Map Discretization for Collision-Free Flights” to compute the set of POIs and corresponding regions (Section 3.1.1), “Routes Computation for Efficient Area Coverage” to compute optimal trajectories for drones within the generated regions (Section 3.1.2), and “Graphical User Interface for Swarm Control and Data Visualization”, which is an operator interface to the system (Section 3.1.3). Next, we summarize these three modules in more details.

#### 3.1.1. Grid-Less Map Discretization for Collision-Free Flights

The “Map Discretization” module calculates the POIs within the ROI in which the UAVs shall gather information. In our setup, we assume that UAVs fly at constant predefined heights. Therefore, the problem can be reduced to finding *I* two-dimensional points ${\vec{p}}_{i}\in R^{2}\;i=1,\ldots ,I$, containing coordinates x,y of the POIs. Given a ROI with a total area $A_{ROI}$, and a sensor that has a footprint $A_{sensor}$, we can calculate the number of POIs *I* as follows:
(1)$$\begin{array}{c} I=\eta \frac{A_{ROI}}{A_{sensor}}, \end{array}$$
where parameter η>0 specifies the desired overlap of sensor footprints.

Our system relies on a proactive collision avoidance. This implies that POIs should be calculated to avoid inter-drone collisions during flight. We solve this by grouping the generated POIs in *K* sets $R_{k},k=1,\ldots ,K$, to which we will refer to as coherently connected regions in the following. Within a region, any two POIs must be reachable without leaving the region. Therefore, as long as each UAV stays in a particular single region, inter-drone collisions are prevented.

To obtain such regions and to avoid collisions with obstacles, while at the same time getting a good coverage of the ROI, the generated POIs must fulfill the following objectives:
POIs within a region should stay grouped together in a coherent fashion.POIs should spread out to cover the whole ROI.POIs should keep distance from obstacles to avoid collisions.POIs should keep distance from POIs that belong to other regions to prevent inter-drone collisions.Drones predefined starting point ${\vec{c}}_{k}$, k=1,…,K must be part of the region.

To model these objectives formally, we define a potential function $\phi_{i}$ for each point ${\vec{p}}_{i}$. This is defined as follows.

Let us assume that the DEM is a grid-based representation of the environment. The center ${\vec{o}}_{i}$ of all grid cells with an elevation higher than some threshold $h_{thresh}$ forms the obstacle set *O*. The combination of all region sets $R_{k}$, k=1,…,K, such that ${⋂}_{k}^{K}R_{k}=⊘$, builds the set $R={⋃}_{k}^{K}R_{k}$. Furthermore, all points $\vec{x}\in R\cup O$ within radius $r_{thresh}$ around ${\vec{p}}_{i}$ are used to define a set $N_{i}=\left\{\vec{x}\in R^{2};∥\vec{x}-{\vec{p}}_{i}∥<r_{thresh}\right\}\setminus \left\{{\vec{p}}_{i}\right\}$. Now we define the potential function $\phi_{i}$ for each POI ${\vec{p}}_{i}$ as:
(2)$$\begin{array}{cc} \phi_{i}= & w_{1}\sum_{{\vec{p}}_{j}\in R_{i}\cap N\left(i\right)}∥{\vec{p}}_{i}-{\vec{p}}_{j}∥+w_{2}\sum_{{\vec{p}}_{j}\in R_{i}\cap N\left(i\right)}\frac{1}{∥{\vec{p}}_{i}-{\vec{p}}_{j}∥}\\ & +w_{3}\sum_{{\vec{o}}_{j}\in O\cap N\left(i\right)}\frac{1}{∥{\vec{p}}_{i}-{\vec{o}}_{j}∥}+w_{4}\sum_{{\vec{p}}_{j}\in \left(R\setminus R_{i}\right)\cap N\left(i\right)}\frac{1}{∥{\vec{p}}_{i}-{\vec{p}}_{j}∥}+w_{5}∥{\vec{p}}_{i}-{\vec{c}}_{k}∥, \end{array}$$
where ${\vec{c}}_{k}$ is the starting point of the region $R_{k}$ such that ${\vec{p}}_{i}\in R_{k}$. Each summand in Equation (Equation 2) corresponds to one of the five objectives listed above. The parameters $w_{i}>0$, i=1,…,5, are introduced to allow an operator to weight the five objectives as desired. The cost in Equation (Equation 2) is constructed such that the derivative of the potential can be interpreted as a force acting on ${\vec{p}}_{i}$. The terms weighted by $w_{2},w_{3},w_{4}$ cause a repulsive force away from the corresponding points ${\vec{p}}_{j}$ ands ${\vec{o}}_{j}$, while the terms weighted by $w_{1},w_{5}$ cause an attractive force towards the corresponding points ${\vec{p}}_{j}$ and ${\vec{c}}_{k}$.

Based on the potential function Equation (Equation 2), we designed an iterative algorithm to calculate the POIs. Initially, the starting point ${\vec{c}}_{k}$ is added to each of the regions. Then, in each iteration we add a new POI to each region until the total number of POIs reaches the predefined limit *I* computed in Equation (Equation 1). POIs are inserted at random locations within the corresponding region. Furthermore, we add a small random noise to POIs locations, which prevents numerical instabilities when two points are too close to each other. Then, for each POI in *R* we calculate the derivative of the potential function with respect to the point’s *x* and *y* coordinates. Based on the derivatives we update the location of each POI in a greedy fashion as follows:
(3)$$\begin{array}{c} {\vec{p}}_{i}\leftarrow {\vec{p}}_{i}+\alpha {\left[\frac{d\phi_{i}}{dx},\frac{d\phi_{i}}{dy}\right]}^{T},i=1\ldots I, \end{array}$$
where α is a parameter that specifies the step size of the algorithm. By iterating Equation (Equation 3) for each point, the POIs "move" in the potential field created by $\phi_{i}$ and arrange themselves such as to minimize Equation (Equation 2) to fulfill optimization objectives. Once the sum of changes of the positions of POIs in two consecutive steps drops below a certain threshold the algorithm terminates. To illustrate how the algorithm works, we depict in Figure 3 three snapshots of the POIs spread.

Given a coherently connected region computed in this way, the next step is to build a graph over the points in the region. To this end, we apply a triangulation algorithm to all points in R∪O after some number of iterations of Equation (Equation 3) (in our implementation this is done after each 100th iteration). Therefore, for each region, we obtain a mesh or graph, which is specified by the POIs in the region and corresponding adjacency matrix $A_{k}$, k=1,…,K that describes connectivity between the points.

Let us point out that the definition of Equation (Equation 2) as a potential function has the disadvantage that there are no guarantees on the fulfillment of objectives 1–5. In fact, we observed that single POIs or small parts of a region’s graph getting disconnected from the rest of a region. Nevertheless, this issue can be easily identified by looking at the singular value decomposition of the Laplacian of the region’s graph. If we detect that a POI or part of the graph is disconnected from the starting point ${\vec{c}}_{k}$ after executing the triangulation algorithm, we delete this part of the graph. The algorithm will then add new POIs until *I* POIs are generated.

#### 3.1.2. Routes Computation for Efficient Area Coverage

Here we summarize how the order in which POIs are visited by the UAVs is computed. Essentially, UAVs should visit POIs efficiently such as to minimize the total traveled distance. This problem can be recognized as a traveling salesman problem (TSP). Classical TSP solvers require a weighting matrix holding the traveling costs—the traveled distance in our case—between all nodes of the graph. Unfortunately, our graphs of POIs are not fully connected. Because regions might be concavely shaped, in a fully connected graph edges might intersect with another region, which might cause possible collisions between routes. Furthermore, obstacles might prevent a direct link between two POIs within a region. Therefore, we must require that UAVs only travel along the edges of the graph provided by the "Map Discretization" module. Thus, if there is not a direct link between any two POIs in the graph, we assume that the UAVs must travel along other edges available in the graph and use other POIs as intermediate stops. This may imply that some POIs will be visited multiple times.

To apply a standard TSP solver, we must calculate the weighting matrix holding the traveling costs, which result from direct and non-direct links between all combinations of POIs in a region. For POIs that are directly linked, it is straight forward and we can use the Euclidean distance. In contrast, for POIs that are not directly linked in the graph, we calculate the distance of the shortest route following the edges of the graph. For the latter, we apply the Floyd–Warshall algorithm [15]. As output of the "Routes Computation" module, we obtain lists of POIs that are optimal traveling routes. These lists are then stored in the DB to be later accessed by drones.

#### 3.1.3. Graphical User Interface for Swarm Control and Data Visualization

The GUI we developed for our system has two key roles. On the one hand, it permits an operator to control the swarm of drones. On the other hand, it visualizes data gathered by drones. We can distinguish between two phases in the GUI workflow: a setup and an online phase.

During setup phase the GUI is used to set input parameter of the "Map Discretization" module (Section 3.1.1). Essentially, the operator can set the ROI, specify the number of POIs and regions, as well as the height threshold $h_{thresh}$ parameter to map obstacles. In addition, the operator can tune the weighting parameters $w_{1}$-$w_{5}$ in (Equation 2) in order to obtain a desired safety distance of POIs to obstacles and other regions, and to obtain a desired POIs density. Furthermore, trajectories output by the "Routes Computation" module (Section 3.1.2) are displayed in different colors (see left-hand side of Figure 4).

During the online phase, our GUI (Figure 4) displays status information provided by drones from their last visit to the “beehive”. The GUI is organized into two main parts: on the left-hand side, it displays a map with trajectories and POIs for each region (differentiable by colors), and the status of POIs represented by a color shade. Light shade means that a POI is unvisited, and dark shade depicts a visited POI. On the right-hand side, the GUI displays the system status on the top part. In particular, it allows an operator to track the following: system run time, percentage of total visited POIs, percentage of visited POIs per region, drone assigned to each specific region, and drones’ connectivity to base station. Furthermore, the GUI displays an overview of (i) drones that are currently active and are gathering information, (ii) spare drones, if any available, which are in the “beehive” waiting to obtain an available route, and (iii) drones that are temporary inoperative; e.g., drones whose battery is being replaced after successfully completing a route.

Additionally, during the online phase, the GUI allows an operator to interact with the system. Specifically, an operator can manually trigger a shutdown procedure for active drones. This is particularly useful if e.g., an emergency landing is required. Moreover, the GUI allows us to retrieve data stored in the DB of a specific POI by simply clicking on it. For example, on the bottom right part of Figure 4, we depict a picture taking by a drone from one of the POIs.

### 3.2. Communication System

Next, we describe the used communication system and the communication concept. Our system relies on a communication link that permits drones to exchange information with the DB at the base station. Information exchange has two key functionalities. First, it allows the base station to assign drones to routes. Second, it permits drones to send collected sensor information to the base station. In particular, drones upload the gathered data (e.g., pictures) in the DB, which is then later visualized in the GUI.

The two aforementioned functionalities specify the communication system requirements, which we summarize next:
The communication range must be in the order of 100 m. Essentially, this range corresponds to the vicinity of the "hive", which is the area in which drones and base station exchange information.The throughput of the communication system shall be sufficient to permit drones to upload gathered information to the DB. Particularly, the exact throughput of the communication system depends on the type of data and the amount of data acquired during the mission.The communication system shall be able to deal with a high package loss and an intermittent communication. As drones fly in and out of the communication range, the system shall be able to deal with these challenges.The communication system physical device shall be light and easily deployable, so that a single operator can transport and rapidly set up the system.

Additionally, for us it is particularly important that the communication system is cost-effective. For a fixed budget, a cost-effective solution allows an operator to increase the number of drones in the system. A higher number of drones speeds up the information-gathering task, as well as it increases the system robustness to eventual single-drone failures.

Based on the aforementioned communication system requirements, we decided to use WiFi technology. The main drawback of using WiFi is the communication range, which is in the order of 100 m. Nevertheless, this is not an issue as our system does not require a larger communication range.

### 3.3. Database System for Multi-Robot Coordination and Information Exchange

The functioning of a multi-agent system heavily depends on the coordination mechanism between different agents. The concept advocated in this work is largely inspired by behavior of bees. In a beehive, the hive itself is the main location where the communication takes place. The implemented DB concept plays a similar role in the proposed system: the coordination of drones, data accumulation and task assignments are all realized through a DB interface, which will be described in the following. Let us point out that although we aim at a system design that is scalable with respect to the number of drones, from a practical perspective the swarm size can consist of 3–10 drones and not 100 of units, as one would expect for an analogy with the bees. There are three main constraints that limit the number of units in the system. First, the communication system has a limited data rate. This is a bottleneck that prohibits e.g., hundreds of drones arriving at the base station simultaneously to upload the collected data. Second, drones require a dedicated landing spot for each drone so that in worst case, all drones can land at the same time. If we had a swarm of hundreds of drones the landing dedicated area shall be prohibitively large. Finally, the logistics and management of large swarms is challenging. Our proposed system can be handled by a single operator who exchanges the batteries of drones that land at the "hive". As soon as the number of units gets too high a single operator cannot deal with it anymore. This would require more operators, which would increase the cost of the system.

The system’s DB consists of three tables that store: (i) information about the drones in the swarm (table “Quads”), (ii) predefined partitioning of the ROI to be explored into sub-regions (table “Regions”), and (iii) the POIs to be visited by the drones (table “Points”). These tables are summarized in Table 1, Table 2 and Table 3. Next we describe in detail the fields that specify the three tables.

Table 1 stores the information related to the drones in the swarm: their unique ID and name, home coordinates of the “beehive” to which the drone is assigned, and a “health” status—a descriptor of a drone’s ability to operate. Home coordinates are unique for each drone; they indicate coordinates where the drone communicates with the database, as well as its take-off and landing position. The Status of the drone identifies the function of the drone within the swarm. Active drones are in the process of gathering information. Drones marked as spare are set in a stand-by modus and wait for new assignments. Drones that are marked as broken are either broken or are currently not registered within the system due to, e.g., a change of batteries.

In Table 2 the information about the regions generated by the “Map Discretization” module (Section 3.1.1) is stored. Apart from the Region_ID and some human-readable naming of the corresponding areal patch, the table also stores information about the location from where exploration of the region should start, and the status of the region. The status of the region can be active, i.e., being currently processed, finished, or spare. The latter is used to designate regions not yet assigned to any drones. The path to traverse each region, generated by the "Routes Computation" module as described in Section 3.1.2, is stored in the designated field PathPlanInfo. It is this last field that is used at later steps to generate entries in Table 3.

Finally, in Table 3 the information about all POIs is stored. Each entry in the table—a POI where to gather information—has specific coordinates, and is associated with a region and a drone. This association takes place as soon as a drone registers itself with the system and changes its status to active. The generated POIs are ordered. This ordering encodes a trajectory, as given by the "Routes Computation" module, which a drone needs to follow: the POIs are visited starting with points with low rank and proceeding to points with a higher rank. Besides the POIs rank, we save the commanded and the actual pose of the drone (coordinates and orientation) at which the measurement was taken, as the latter might slightly deviate from the “commanded” pose. In addition, Table 3 also indicates which POIs were visited by drones using the status field. When drones request assignments from the DB, only the unvisited POIs from the region are assigned to the drone. Finally, for each region the “Points” table also stores the time-stamped information collected at visited POIs such as e.g., imagery data.

## 4. System Components—Drones

Drones play the role of sensor-carrying platforms that collect information and “physically” bring the data to the base station. Naturally, drones require sensors to collect the information of interest, a communication system to transfer the collected information, and some onboard computer that allows for autonomous waypoint-based flight. Next we summarize how we realize these components in our system.

### 4.1. Sensors for Information Gathering and Drones Positioning

Drones typically build on multiple sensors to fly autonomously. In particular, here we focus on the two sensors that are the most relevant for our system realization: sensors for information gathering, and sensors for positioning. For information gathering, we designed our system so that it permits an operator to use a wide range of sensors, depending on the application of interest. In this sense, we designed the system and, in particular, the DB so that measurements output from a wide variety of sensors can be stored.

In addition to gathering information, drones heavily rely on a positioning sensor to fly autonomously and to geo-reference the gathered data. In our system, we require drones to be equipped with a GPS receiver to continuously position the drone. In this respect, a standard GPS receiver (approx. 5m of accuracy), is sufficient to position the drone.

### 4.2. Communication System

The communication system is a fundamental module in our system design. Essentially, it is the mean that drones must communicate with the base station. In terms of the requirements needed for the communication system, these are equal as the ones that we specified for its base station counterpart (see Section 3.2).

### 4.3. Onboard Computer for Autonomous Information Gathering

The onboard computer is the drone’s “brain”. It is responsible for running the algorithm that allows a drone to gather information autonomously, and to transmit the information to the base station. In Figure 5, we summarize in a block diagram the information-gathering algorithm, which we explain next in detail.

Once a UAV is activated, it requests mission data from the base station. Please note that this block is colored white in Figure 5 to symbolize the algorithm steps that require a connection between drone and base station. Mission data is stored in the DB. It contains: a unique region, a flying trajectory, and the information of the already visited POIs in this region. We remind that regions are output from the "Map Discretization" module (Section 3.1.1), while the trajectory is the output of the "Routes Computation" module (Section 3.1.2). Trajectories are generated such that collisions with obstacles are prevented, and flight in other regions is prohibited.

After receiving the mission data, the UAV sends the *take-off* signal to the DB and soars vertically to a predefined height, which is set individually for each mission. Once the drone reaches the desired flying height, the autonomous information-gathering workflow starts. This is highlighted in orange color in Figure 5. Following the beehive analogy, the UAV follows commands from the hive. Hence, if there are still unvisited POIs and if the battery voltage is sufficient, the UAV approaches the next POI to take measurements. At this stage, it is very likely that the UAV disconnects from the DB due to the potential large distance between the UAV and the base station. In this situation the UAV only relies on the obtained mission data and its onboard sensors.

Once all POIs are visited or the battery voltage is too low to continue the mission, the UAV returns to its home position following the shortest route. To calculate this route to the home position the UAV uses A* algorithm [16]. As input, A* takes the graph that corresponds to the drone’s assigned region. After reaching the home position, the UAV establishes a connection to the DB and transmits the gathered measurements.

## 5. System Analysis

We analyze here the system’s performance in simulations as we increase the number of UAVs. In particular, we measure the time that the system requires to complete its mission: covering all POIs within a predefined ROI. For all simulations, we assumed a ROI of a fixed size. The ROI size is given by the area that can be covered by 8 UAV flights, where the duration of a single flight is 12 min. Please note that we choose this value because it is the one that we later use for the experimental evaluation.

Here we compare two different scenarios. In the first scenario, we divide the ROI in 8 regions, and increase the number of UAVs from 1 up to 8. In addition, we analyze the effect in the system performance of the time required to exchange drones batteries and to re-launch the system. We call this *setup time*. For simplicity, we assume that the setup time is constant during an experiment and independent of the number of drones. In Figure 6a, we depict the simulation results for this first scenario. We can observe that the total mission time remains constant for 4–7 drones systems. This results from the fact that 4 UAVs can cover the ROI in two rounds with four simultaneously flying UAVs. Instead, 6 UAVs would require one round with six simultaneously flying UAVs, and an additional round with two simultaneously flying UAVs. This sums up two flying rounds, which is the same as for the 4 UAVs system.

In a second scenario we divide the ROI into three regions. The first two regions have equal size and can be covered with 3 flights, while the third one can be covered with 2 flights. Figure 6b depicts the simulation results for the second scenario. We can see that in the absence of setup time, the system’s performance does not increase for several UAVs larger than 3. In contrast, as we increase the setup time we can observe that the performance increases for a system with up to 5 drones until it remains constant.

Simulations results allows us to understand the effect of the setup time and of the number of UAVs in the system’s performance. Based on these results, and taking into account the system’s performance and cost, we decided to use 4 drones for the experimental evaluation.

## 6. System Implementation

We described in the previous sections our algorithm workflow, and the different components that are part of our system design. Here we specify the specific software and hardware components that we used to implement and evaluate our system. Please note that these components are merely an example of possible components, but different ones could be used to realize our proposed system.

### 6.1. Software

The code developed for the system is written in Python. We used multiple libraries to program the modules of our system. Specifically,

*Drones Monitoring and Data Visualization*: this module is implemented using PyQT libraries to develop the GUI.*Map Discretization*: this module requires a DEM of the terrain to prevent drones’ collisions with obstacles. Here we used a DEM that was computed following the methodology described in [17].*Routes Computation*: routes are computed from the POIs graph by solving a TSP. For that, we use the Floyd–Warshall algorithm [15] contained in the open source OR-Tools [18] library. The OR-Tools library is developed and maintained by Google, and provides heuristics and metaheuristics solutions to solve optimization problems.*Database*: the DB is implemented using a MySQL server architecture that is accessed by the drones directly through a classical TCP/IP protocol. This setting also permits accessing the server from a remote location. For instance, if the base station communication system is connected to internet, e.g., using 4G interface, the data collected on the field can be accessed from a remote control/coordination center by directly connecting to the DB. We used the Python library *mysql-connector* to connect, read and write from the DB.*Onboard Computer Algorithm*. The onboard computer runs the algorithm for autonomous information gathering. This algorithm must interact with the drone’s GPS sensor and flight controller to command waypoints to which the drone shall fly. To facilitate the interaction of the information-gathering algorithm with the drone’s sensors and flight controller, we use the robot operating system (ROS).

### 6.2. Hardware

Our system is composed by multiple hardware modules, which cover a large range of functionalities. Instead of specifying the implementation details of every single module, we focus on the ones that from our perspective, are the most relevant for the reader to understand our system. These are the following:
*Communication System*. Based on the communication system requirements, specified in Section 3.2 and Section 4.2, we decided to use WiFi technology. On the base station side, this permits us to use a standard WiFi router, which is lightweight and cost-efficient. On the drones side, we decided to use a WiFi dongle for the same reasons.*Database*. The required performance of the DB is essentially determined by the number of used swarm elements and amount of data collected by drones. For instance, in our experiments we used a small swarm size (<5 units) and collected imagery data. For these purposes, a low-performance server running on a small Raspberry Pi (RPi) computer was completely adequate. Higher performance systems can be deployed when more drones are used or when large amounts of data (e.g., saved video stream) are stored and accessed by the system.*Drone platform*. We used Astec Hummingbird quadcopters. These are light UAVs that have a maximum take-off weight of 710 g. Taking into account the platform’s weight, this leaves us 200 g to include additional payload. In particular, we include the following payload: an onboard computer, a WiFi dongle, and an information-gathering sensor. We would like to remark that our drone platform could easily be exchanged for any custom-built or off-the-shell drone that satisfies the minimal hardware requirements specified in Section 4.*Onboard Computer*. We use a RPi 2 as onboard computer. A RPi is a light computer that can be easily carried by an Astec Hummingbird. The RPi is connected to the UAV flight controller through a serial communication interface. This is used by the information-gathering algorithm to send waypoints to which the drone shall fly to. Additionally, the flight controller reads information from the GPS receiver and an inertial measurement unit (IMU). This information is then forwarded to the RPi to compute drone’s pose.*Sensor*. Here we use an imaging sensor—visual camera—to gather information. In particular, we use a RPi Camera Module V1. This is a consumer level camera that weights only 3 g. This low weight is a very desirable characteristic, as it impacts directly the flight time of the UAV. The sensor of the camera is an OmniVision OV5647 and has a resolution of 2592×1944 pixels. We placed the camera below the UAV pointing down. The main drawback of the camera is the use of a rolling shutter. This can cause noise in pictures if the UAV is not stable enough. Nevertheless, we would like to point out that a global shutter camera can likewise be used. Moreover, we could also use any other sensor by introducing only minor modifications in our system.

## 7. Experimental Evaluation

### 7.1. Setup

We carried out 4 field experiments (*Experiments 1–4*) to evaluate our system. Experiments took place in our experimental area, located in the vicinity of Poecking, Bavaria, Germany. The area measures approx. 200×200 square meters, and is depicted in Figure 7a.

For all experiments, we divided the ROI in 3 regions, and generated 300 POIs. Drones flew at a height of 10 m. The drones used for the experiments were the Astec Hummingbirds (see Figure 7c). We used systems consisting of several drones ranging between 1 and 4. We can see in Figure 7b an example of three drones flying while they take images of a ROI. In the following sections, we refer to each of the drones by its name, which is the following: Hans, Orville, Charles, and Otto. Next, we present the experimental results of our tests.

### 7.2. Results

The main objective of our system is to gather information in all POIs that are part of the ROI. POIs are organized in different regions, and for each region a flying route is computed and stored in the DB. Essentially, we want drones to fly autonomously to complete routes without missing any POI.

To verify whether drones gathered information at all POIs, we plot in Figure 8 the nominal and actual trajectories of the drones. In particular, Figure 8 depicts trajectories that correspond to *Experiment 3*, in which three regions and three drones were used. In Figure 8a we show the trajectories as they were saved in the DB, while in Figure 8b we depict the trajectories as they were recorded by each drone’s GPS receiver. First thing that we can observe is that all POIs were measured, which indicates that drones were able to complete the information-gathering task. Second, we note that each region was assigned multiple times to different drones. As the regions are large, they cannot be covered by a single-drone flight. Our system design deals with this limitation, and assigns the region to a spare drone while the battery of the previous one is being replaced.

We can also see in Figure 8 that drones nominal and actual positions do not match perfectly. Essentially, this is because we assume a UAV reaches a POI once it is within 1 m radius from it. Let us also highlight how the routes for returning to home are constructed. In most of the cases, we can observe that these routes are shorter than the routes computed to gather information. This is because these routes are computed using A* algorithm with the goal to return to home following the optimal (in terms of traveled distance) path.

In addition, to achieving a full coverage of the ROI, we design our system (i) to be scalable with respect to the number of UAVs in the system; and (ii) to be robust against potential drones failures. In the following, we analyze these two aspects in more detail.

#### 7.2.1. Scalability with Respect to the Number of Drones

We carried out three different experiments (*Experiments 1, 2 and 3*) in which we fixed the number of regions to 3, and we varied the number of drones between 2 and 4. First, we evaluated the time required to cover a certain area as we increase the number of drones. Here we compute the covered area as the number of square meters on the surface observed with the drones cameras. The intended overlap of the images has been taken into account for calculating the square meters.

In Figure 9a we show the time required to cover a certain area for each of the experiments. Please note that the gaps in the thick lines correspond to the instants in which drones batteries are being changed and, therefore, no area is covered. In addition, we draw a regression line for each of the experiments, together with the regression coefficient. The coefficient represents the speed of the experiment, where a smaller coefficient corresponds to a shorter experiment time. This means that the experiment with two UAVs was completed with a speed of 4 $\frac{m^{2}}{s}$. The other two experiments have almost the same coefficient, which corresponds to a speed of approx. 8.3 $\frac{m^{2}}{s}$. This tells us that having several drones (4 drones) larger than the number of regions (3 regions), i.e., having a spare drone, does not bring a higher performance to the system in terms of the covered area.

Second, we analyzed the time that each of the drones in the system needs to travel a certain distance. Results are depicted in Figure 9b, where the color of the plots refer to different experiments. In addition, we draw a rectangle that represents the area that is created by taking the maximum value of the traveled distance and time. A smaller area represents a more effective UAV allocation. In this case, we can observe that the area of the rectangles is reduced as we have more drones in the system. We can also see that adding a third UAV brings a high performance increase. In contrast, adding a fourth UAV only leads to a slight improvement.

#### 7.2.2. System Robustness Against Drones Failures

Our proposed system can cope with a drone failure (e.g., a communication outage or, more drastically, a drone crash) and can still complete the information-gathering task. We simulated a drone failure in *Experiment 4* by landing the UAV Otto during flight. Figure 10 shows this scenario, where Otto flies in the middle right part of the ROI (see the red curve in Figure 10b). Its failure is simulated approximately at the location (160 m, 140 m). Since Otto is not able to return home in a predefined timeout, the system that runs on the base station declares it as broken. Hence, its trajectory is not shown in Figure 10a, which shows the trajectories in the DB. Although Otto was not able to complete its mission, we can observe in Figure 10a that Charles and Hans took over its POIs, and covered them all.

In addition to the drones trajectories, we plotted in Figure 11 the assignment of the regions. Figure 11a, which corresponds to *Experiment 3*, shows the ideal case, where there is no failure. There we can observe how each of the regions is assigned to different drones, while we change drones batteries. In *Experiment 4* we simulated a drone crash (see Figure 11b). We can see that Otto starts its mission. Then it crashed and Charles, followed by Hans, took over Otto’s region. Results of this last experiment exemplify the capability of our system design to cope with drones failures.

## 8. Conclusions

In this paper, we proposed and experimentally validated the design of a multi-agent system for autonomous information collection to be used in emergency response scenarios. The designed system has been tested with multirotor UAVs for information collection, but fixed-wing drones can be used as well with very little modifications. The system was specifically designed to be cost-efficient and simple; at the same time to scale with the number of used drones and to tolerate a loss of multiple drones without causing a total mission failure.

This was achieved by sacrificing the drone-to-drone communication or mesh networking, along with reactive collision avoidance. Instead, by using analogy with bees, a centralized communication and planned drones trajectories were centrally generated to avoid collisions with obstacles or other drones. To this end, we proposed in this work a map discretization algorithm that generates collision-free trajectories. The generated trajectories were then communicated to drones using simple WiFi links. As such, the drones can merely traverse the pre-planned trajectories, taking sensor measurements to physically bring measurement data to a central “beehive”. To plan the trajectories, the whole exploration area was partitioned in several non-overlapping regions, where drones can fly without collision. The number of regions, their size, as well as the number and density of waypoints within the region are sensor and application specific parameters. The choices made in the paper are to demonstrate the performance of the whole system in the experiments. They must be selected, however, depending on the specific sensor, type of the used drone, and type of collected information. However, experiments have shown that the number of regions should be smaller or equal to the number of drones used in the system: in this case the exploration speed is higher, and spare drones can be used in case of drone failures. Additionally, let us stress that the absence of peer-to-peer data communication links implies significantly reduced payload; this simplifies requirements on the size of the drones. Furthermore, pre-planned trajectories alleviate the need for an onboard sensor and computer for reactive collision avoidance. This likewise reduces the payload weight. Altogether, lighter and cheaper drones can be used. The consequence of using lighter and cheaper drones is the fact that a loss of a single drone in a system can be tolerated more easily.

The proposed design also proved experimentally to be robust against latter events. In particular, when a drone does not return to the control station after a predefined timeout, its tasks can be reassigned either to spare drones (those that are not actively collecting information), or to other drones used in the system. This ensures that data will be continuously collected, even if only one drone is used in the system.

Finally, we believe that the proposed design is simple yet effective for autonomous information gathering in very diverse applications, where real-time data collection or surveillance is not required. The implemented concept can be extended to different types of drones, sensors and applications with very limited modifications of the whole systems, making it an attractive basis platform for further extensions. A promising future extension is the formulation of the information-gathering problem as a FANET. This formulation would allow us to extend our system to real-time data collection applications.

## Figures and Tables

**Figure 1 sensors-19-04349-f001:**
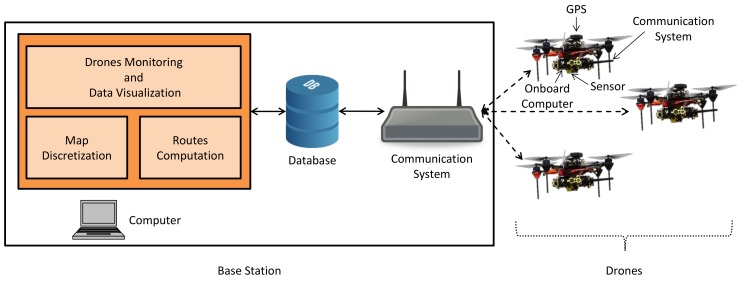
System block diagram.

**Figure 2 sensors-19-04349-f002:**
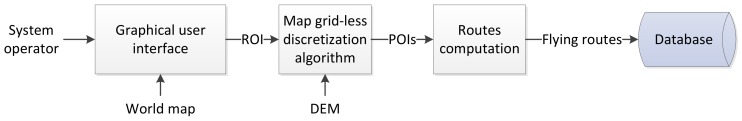
Base station workflow.

**Figure 3 sensors-19-04349-f003:**
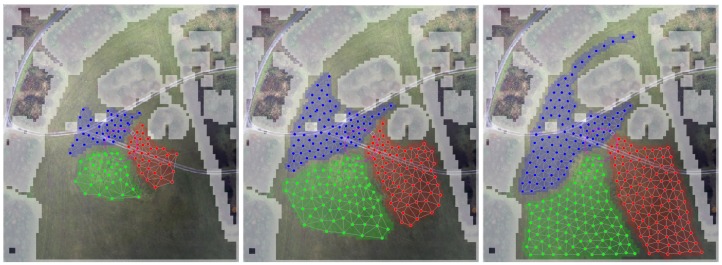
Three snapshots of the map discretization algorithm for partitioning the ROI into three regions (red, blue, green). The green circumference delimits the drones starting position. Obstacles are marked in gray.

**Figure 4 sensors-19-04349-f004:**
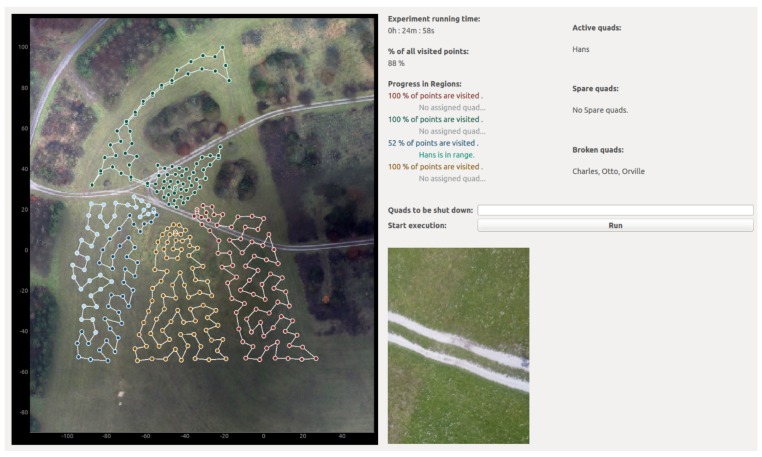
GUI for swarm control and data visualization.

**Figure 5 sensors-19-04349-f005:**
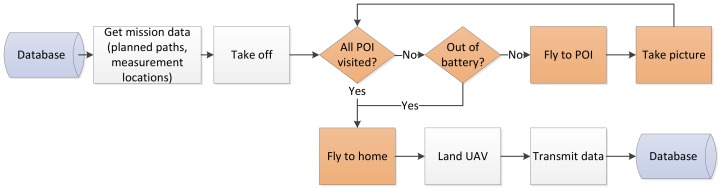
Algorithm for autonomous information gathering. Orange boxes indicate algorithm steps in which the UAV is gathering information and is, therefore, not connected to the base station. White boxes indicate steps in which the UAV shall be connected to the base station.

**Figure 6 sensors-19-04349-f006:**
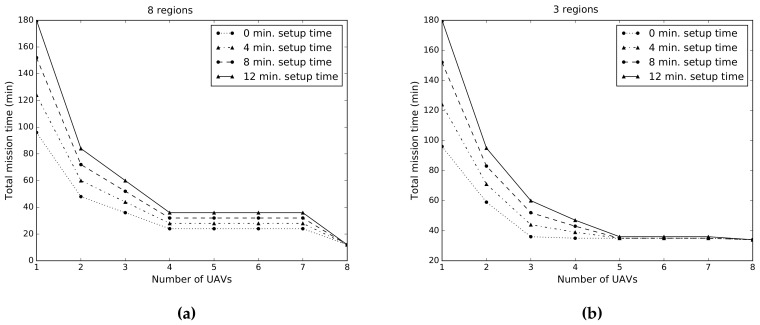
Simulation results that evaluate the total mission time in terms of the number of UAVs in the system, and of the system’s setup time. For all simulations we assume a ROI that has a constant size. (**a**) We divide the ROI in 8 regions. (**b**) We divide the ROI in 3 regions.

**Figure 7 sensors-19-04349-f007:**
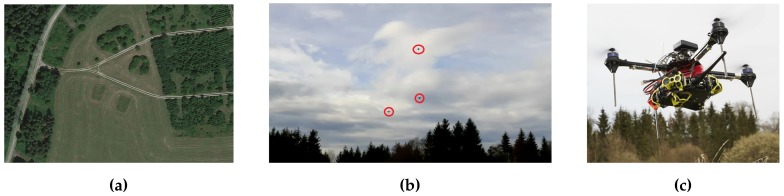
Experimental setup to evaluate our system design. (**a**) Aerial image of the area in which we carried out the experiments. (**b**) Three drones flying while they take images of a ROI. (**c**) One of the Astec Hummingbird quadcopters that we used for the experiments.

**Figure 8 sensors-19-04349-f008:**
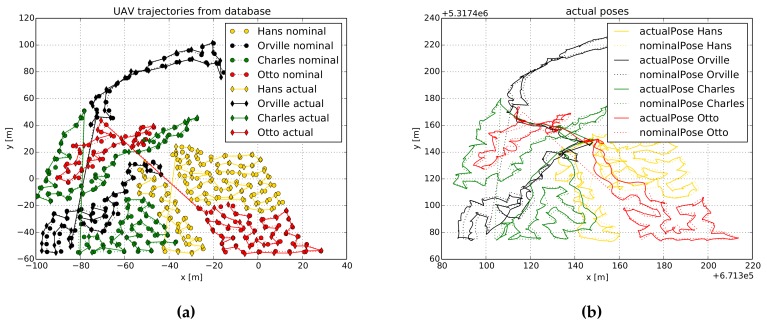
Nominal and actual trajectories of a system composed by 3 drones and 3 regions. (**a**) Trajectories as stored in the DB. (**b**) Trajectories recorded by each drone’s GPS receiver. In (**a**) we plot trajectories in a local coordinate frame, while in (**b**) we plot them in a global coordinate frame.

**Figure 9 sensors-19-04349-f009:**
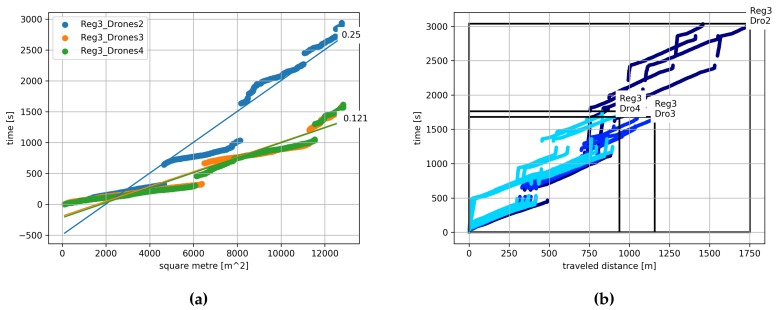
Algorithm scalability with respect to the number of drones. We evaluate a system of 2, 3 and 4 drones. (**a**) Time needed to observe a certain number of square meters. (**b**) Time required for each UAV to travel a certain distance.

**Figure 10 sensors-19-04349-f010:**
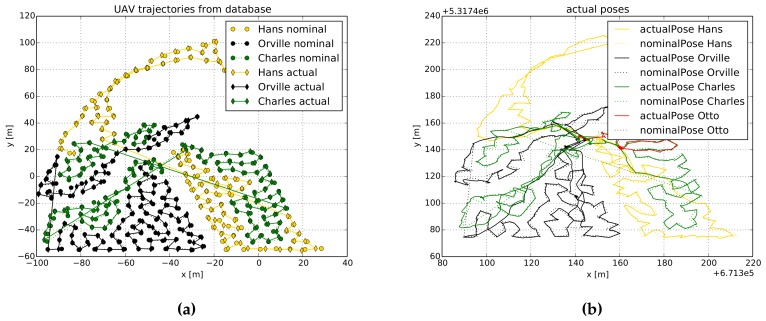
Nominal and actual trajectories of a system composed by 4 drones. (**a**) Trajectories as stored in the DB. (**b**) Trajectories recorded by each drone’s GPS receiver. In (**a**) we plot trajectories in a local coordinate frame, while in (**b**) we plot them in a global coordinate frame. Here we can observe that Otto crashed, as it was not able to complete the route. Charles, followed by Hans, took over Otto’s route, which demonstrates the system’s robustness against a drone crash.

**Figure 11 sensors-19-04349-f011:**
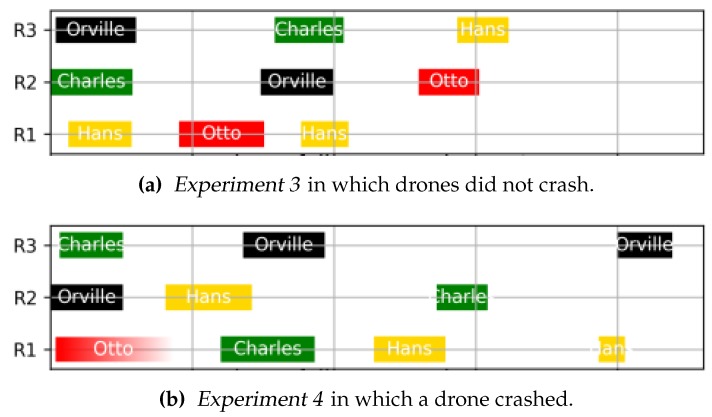
Assignment of regions for three different experiments. For each of the drones we represent the instant of time in which they were assigned to a region (R1, R2, or R3). We depict results for one experiment that was successfully completed (**a**), and for one experiment in which a drone crashed (**b**).

**Table 1 sensors-19-04349-t001:** Definition of the table “Quads”.

Table Field	Field Description
Quad_ID (key)	ID of the quad.
QUAD_NAME	Human-readable name of the quad.
Xcoord_HOME	X coordinate of the quad home position.
Ycoord_HOME	Y coordinate of the quad home position.
Zcoord_HOME	Z coordinate of the quad home position.
Status	Status of the drone. This field can take only 3 possible values: active, broken, or spare.

**Table 2 sensors-19-04349-t002:** Definition of the table “Regions”.

Table Field	Field Description
Region_ID (key)	ID of the region for inspection.
Region_NAME	Human-readable name of the region.
Xcoord_START	X coordinate of the drone starting position.
Ycoord_START	Y coordinate of the drone starting position.
Zcoord_START	Z coordinate of the drone starting position.
Status	Status of the region. This field can take only 3 possible values: finished, active (currently being explored), and spare (waiting to be assigned).
PathPlanInfo	Binary object that stores the graph generated by the “Routes Computation” module.

**Table 3 sensors-19-04349-t003:** Definition of the table “Points”.

Table Field	Field Description
Point_ID (key)	ID of a POI for inspection.
Region_ID	ID of a region to which the POI is assigned.
Quad_ID	ID of a quad to which the POI is assigned.
Point_Rank	Rank of a POI defined by the "Routes Computation" solution.
Xcoord	X coordinate of the POI.
Ycoord	Y coordinate of the POI.
Zcoord	Z coordinate of the POI.
Xcoord_Actual	Actual X coordinate of the drone when it visited the POI.
Ycoord_Actual	Actual Y coordinate of the drone when it visited the POI.
Zcoord_Actual	Actual Z coordinate of the drone when it visited the POI.
Roll_Actual	Roll of a drone at the POI when it was visited.
Pitch_Actual	Pitch of a drone at the POI when it was visited.
Yaw_Actual	Yaw of a drone at the POI when it was visited.
Status	Status of the POI. This field can take only two possible values: visited, and unvisited.
Image	Actual image taken at the POI.
Image_TIME	Date and time stamp of an image taken at the POI.

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
