# Peer review of "Beehive-Inspired Information Gathering with a Swarm of Autonomous Drones"

_sensors, 2019, doi:10.3390/s19194349_

Round 1
Reviewer 1 Report
A FANET approach based on bees behaviour is presented. The proposal is really interesting, however it is not clear if the technique is effectiveness.
Authors did not compare their technique with other techniques. Morever, just a test based on 4 drones is presented. Maybe, it was interesting to perform simulations varying the number of nodes and comparing it with other techniques.
Another problem is that network metrics are not defined. In order to prove that the technique has a good performance, network metrics must be quantified.
How the FANET will perform if more drones were added? The proposed technique can really scale?
Finally, system workflows (section 2) should be better understood by algorithms.
I strongly recommed more tests or simulations to prove the effectiveness of the proposed algorithm.
Author Response
A FANET approach based on bees’ behavior is presented. The proposal is really interesting, however it is not clear if the technique is effectiveness.
First of all, we would like to thank the reviewer for the time devoted to our work, and for the constructive comments.
The reviewer frames our work within the topic of FANETs. At this stage, we would like to clarify the difference between our work and the classical definition of FANETs. A FANET is typically studied in the context of communications and refers to a flying ad-hoc network. Typically, research on FANETs focuses on the communication aspect of flying networks. In contrast, our work does not focus on communication. Instead, we propose a solution that permits us to focus on the information gathering aspect while leaving the communication problem as “solved”. This is essentially the reason why we propose a bee-hive concept in which communication is centralized. This permits us to use affordable off-the-shelf communication devices. Additionally, our approach does not require the development of algorithms to solve the communication issue.
Of course, we consider that formulating our problem as a FANET will enhance our system. This is definitely one of the directions of future work to further develop our algorithm. In order to highlight the relationship between FANETs and our work, we included some text in the paper’s introduction and conclusion.
Second, we would like to comment on the effectiveness of the technique. We proved in experiments the effectiveness of the system in multiple experiments with different number of drones. Additionally, we followed reviewer’s suggestions and included a new section that evaluates the proposed technique in simulations. In our next answers, we elaborate on the simulations we carried out.
Authors did not compare their technique with other techniques. Moreover, just a test based on 4 drones is presented. Maybe, it was interesting to perform simulations varying the number of nodes and comparing it with other techniques.
Thank you for this comment. First we would like to justify why we did not compare our technique with another ones. The literature research revealed that there is no system in the literature that solves our problem: gathering information with a swarm of affordable drones, and demonstrate it in field experiments. This we pointed out in section 1. Therefore, it was not possible for us to carry out a comparison. Nevertheless, we understand a single-drone system as a benchmark for information gathering tasks, as this is the most commonly used solution. In this respect, we show in the paper how our system composed of multiple drones outperforms its single-drone counterpart.
We agree with the reviewer that it would be interesting to analyze the system performance with more than 4 drones. To this end, we added a new section to the paper (Sec. 5) and analyzed the system performance in simulations with up to 8 UAVs. In addition, we also evaluated the influence of the system setup time on the overall performance.
We are now strongly convinced that combination of simulations plus multiple experiments allows us to draw definitive conclusions about our proposed design.
Another problem is that network metrics are not defined. In order to prove that the technique has a good performance, network metrics must be quantified.
Thank you very much for this clarification. As we pointed out in our first answer, our paper’s goal is on the information gathering aspect and not on communications. Therefore, we focused the evaluation of our system in terms of information gathering. This is why we evaluated the system in Secs. 5, 7 in terms of total mission time, area coverage and distance travelled by drones.
Since our focus is not on communications, we believe that analyzing network metrics would be misleading for the reader. In any case, we fully agree with the reviewer that the analysis of network metrics is definitely an aspect that we should be considered in a FANET formulation of our problem.
We modified the conclusion of the paper to highlight that our system could be also formulated as a FANET.
How the FANET will perform if more drones were added? The proposed technique can really scale?
Thanks for the valuable comment. We would like to split this answer in two parts. First, we comment on how the system scales in terms of the information gathering efficiency. In this case, we focus on the information aspect, which is our goal in the paper, and we consider communication as perfect. Under these assumptions, we carried out multiple simulations to analyze the system scalability. To this end, we added a new section to the paper (Sec. 5) with this evaluation, which shows that the system scales with the number of drones.
However, although communication is not the main focus of our work, it is true that is a fundamental aspect as it is been remarked by the reviewer. Therefore, we would like to comment next on our current system limitations in terms of communication. In our system we are using a standard Wi-Fi infrastructure without ad hoc capabilities, therefore, the performance with the number of drones is related to the limitations of the standard. These limitations are not only related to the number of drones, but also to other factors like the interferences with other devices working on the same frequency, separation between the UAVs, multipath components and the throughput needed by each UAV. Therefore, it is really challenging to select a scenario to estimate how the performance will decrease. One of the main problems we can observe on using Wi-Fi with an increasing number of drones is that a bottleneck will be created if a large number of them want to send the collected information at the same time. Coming to the second question, the bottleneck problem is not a limiting factor of our technique and is due to the limitations of Wi-Fi. In our approach, we need to be in contact with the central station in order to send the information and, for Wi-Fi the main limitation is the range. Using other communication protocols this range can be extended, for example, using higher transmits power or using relays nodes, therefore, two problems can be solved, (i) the data can arrive faster, and (ii) the drones will transmit the information just after is measured, avoiding the need of high throughputs when several drones are close to the “hive”.
In response to this comment, we added a paragraph to section 3.3 to elaborate more on the scalability and the limitations of our current version of the system. In addition, we added some text to the introduction and conclusion to highlight how our system could benefit from a FANET formulation of the problem.
Finally, system workflows (section 2) should be better understood by algorithms.
Thank you very much for this comment. We fully agree with the reviewer that explaining the system workflow as an algorithm would ease the understanding of the system.
For the case of the drones system, we already had an algorithm block diagram (Fig. 5) that explains the workflow. This was unfortunately not the case for the base station. Therefore, we have now created a block diagram to better explain the base station workflow (see Fig. 2 in the current version of the manuscript).
We really hope that now the system workflows are clear to the reader.
I strongly recommend more tests or simulations to prove the effectiveness of the proposed algorithm.
As we previously pointed out, we added a new section to the paper (Sec. 5) that analyzes the system performance in simulations.
At this point, we would also like to remark the importance of this paper’s experimental results. In contrast to most of the works in the literature, in which a single experiment is carried out, here we realized 4 field experiments.
Reviewer 2 Report
First of all, your paper is nicely organized and you presented the operation of your system in a way that is easy to follow. Your overall system framework is nicely structure. Your work addresses a simple solution for a complex problem. Your software and hardware development are properly explained. Your overall experiments are properly documented and they have consistency.
However, my biggest issue is the overall application of your system. You specify its application into facilitating emergency responders activities or mission. One thing that is not clear is how long it takes for the overall process to get the “nectar” to be analyze in the “beehive”. For rescue missions in which every second counts to save a life, your system takes to long to get data to the rescue workers. For such a critical application, almost-real time data must be delivered to the rescue workers. I see your concept as a great proof of concept of centralized multi-agent coordination. Your system is very relevant for survey related operations, maybe package delivery, or any activity that does not get affected by any delay on data to arrive to the user. I like the idea of removing a dependency of drone to drone communication on your particular system.
I recommend you rethink the objective of your system and evaluate the level of urgency for the data to arrive. Also, your overall prototype system is not really cost effective. At the moment that I read that you use an AsTec robot I saw an expensive price tag of a system I do not want to afford to lose. By enabling predefined waypoints and points of interest your system focusses on self-awareness of the drone (IMU, GPS, etc.). However, I would not like to lose a 3,515 euro AsTec Hummingbird that is not longer being manufactured since Intel took over. You would really proof a cost effective system is you manufacture your own drone. A good small drone can be built with about 350 to 400 dollars. Or you can get a DJI Mavic for 700 to 800 dollars and add external sensing with a Raspberry Pi. So it is still essential to look into the situational awareness details of your drone (lidar, radar, collision avoidance).
With what you develop you can specify assumptions or boundary conditions. Your system is trying to solve a piece in the puzzle, but leaving many others in the open. Just be clear of your constraints and limitations. There are weather conditions, animals, projectiles among other things that always exist within the airspace that have to be account for. Even if the path planning seems to be optimal for drone to drone self-awareness within the same flock, on your paper you need to be clear that you understand about the other issues. It is welcome to specify your scope and letting the reader know that you aware that your approach deals with a solution that will need to further complemented by other subsystems.
Your work is good, your algorithms make sense, and your team were successful into implementing an experimental situation to test your approach.
In summary work on the following points:
*Define a realistic scope for your system, in a rescue mission your system will take to long to deliver critical data about victims. LoRa can go beyond WiFi and can transmit that data from your drone.
*Specify that you might need to create a new prototype with a cheaper drone to truly test it to be cost effective. Right know you worked with ideal conditions with a top of the line multirotor. Take care of those Hummingbirds.
*Your POI and ROI path planning still not guarantee immediate safer path but is a good work in progress. I am glad you were honest about it. That approach can be a great application into and Unmanned Traffic System that can be implemented for a dynamic airspace or airway cell risk evaluator.
Author Response
First of all, your paper is nicely organized and you presented the operation of your system in a way that is easy to follow. Your overall system framework is nicely structure. Your work addresses a simple solution for a complex problem. Your software and hardware development are properly explained. Your overall experiments are properly documented and they have consistency.
We would like to thank the reviewer for the time devoted to our work, and, of course, for the positive summary. Next we address each of the critic points raised by the reviewer.
However, my biggest issue is the overall application of your system. You specify its application into facilitating emergency responders activities or mission. One thing that is not clear is how long it takes for the overall process to get the “nectar” to be analyze in the “beehive”. For rescue missions in which every second counts to save a life, your system takes to long to get data to the rescue workers. For such a critical application, almost-real time data must be delivered to the rescue workers. I see your concept as a great proof of concept of centralized multi-agent coordination. Your system is very relevant for survey related operations, maybe package delivery, or any activity that does not get affected by any delay on data to arrive to the user. I like the idea of removing a dependency of drone to drone communication on your particular system.
I recommend you rethink the objective of your system and evaluate the level of urgency for the data to arrive.
We agree with the reviewer; responsiveness of the system is limited due to the chosen communication solution. By building upon the analogy the bees, it is clear that nectar is not collected by bees instantaneously. Similarly without system, the irreducible delay is introduced by the time a drone requires to comeback from the “field” to deliver the alarm signal.
As such, the proposed solution in the form discussed in the paper is better adapted to scenarios that are less time critical.
In the new version of the manuscript we introduced corresponding text to explicitly point this out.
Also, your overall prototype system is not really cost effective. At the moment that I read that you use an AsTec robot I saw an expensive price tag of a system I do not want to afford to lose. By enabling predefined waypoints and points of interest your system focusses on self-awareness of the drone (IMU, GPS, etc.). However, I would not like to lose a 3,515 euro AsTec Hummingbird that is not longer being manufactured since Intel took over. You would really proof a cost effective system is you manufacture your own drone. A good small drone can be built with about 350 to 400 dollars. Or you can get a DJI Mavic for 700 to 800 dollars and add external sensing with a Raspberry Pi. So it is still essential to look into the situational awareness details of your drone (lidar, radar, collision avoidance).
With what you develop you can specify assumptions or boundary conditions. Your system is trying to solve a piece in the puzzle, but leaving many others in the open. Just be clear of your constraints and limitations. There are weather conditions, animals, projectiles among other things that always exist within the airspace that have to be account for. Even if the path planning seems to be optimal for drone to drone self-awareness within the same flock, on your paper you need to be clear that you understand about the other issues. It is welcome to specify your scope and letting the reader know that you aware that your approach deals with a solution that will need to further complemented by other subsystems.
Thank you for the valuable comments. First, we will comment on the issues raised regarding cost-effectiveness. It is a very relevant point that cost of the system can be drastically reduced by using cheaper version of drones, instead of ~3,5k per piece for AscTec Hummingbirds. In fact, choosing AscTec Hummingbirds is motivated by the fact that these were the drones that we had at our disposal for the project. However, we would like to remark that exchanging our drones with a cheaper prototype that satisfy minimal hardware component requirements is possible without any effect on the system design as addressed in Sec. 4. Actually, as we can see in the paper structure, the concrete drones used are not specified till Sec. 6.2. In fact, for our system design we only require that drones are equipped with:
an onboard computer that incorporates an autopilot functionality to fly to a POI autonomously. an accurate and reliable localization module, i.e. global positioning system (GPS) module. an appropriate sensor stack for information gathering (application based: visual cameras, lidar, etc.) a communication system to exchange information with the database.We address reviewer’s comment by adding some text in 6.2.
Next we will comment on the second issue raised by the reviewer: assumptions that we do on the system’s operational conditions. We fully agree with the reviewer that this aspect was not properly tackled in the paper.
Of course, we are aware that our system works under the following conditions - (i) drones know where they are thanks to GPS, (ii) accurate information about static obstacles is present, (iii) there are no dynamic obstacles during algorithm execution. In case some of these conditions do not hold, our system would need to rely on external sensors and reactive planning algorithms (as pointed out by the reviewer). This was in fact out of the scope of our work, but was not properly addressed in the paper. Therefore, we now included some text in the introduction to clearly specify the aforementioned constraints of the system.
Your work is good, your algorithms make sense, and your team were successful into implementing an experimental situation to test your approach.
Thank you very much for the positive appreciation.
In summary work on the following points:
*Define a realistic scope for your system, in a rescue mission your system will take too long to deliver critical data about victims. LoRa can go beyond WiFi and can transmit that data from your drone.
In connection to the comments made above, we absolutely agree that the reviewer presents a valid case. It is indeed possible to have a low-cost, long-range and low-bit-rate systems, like LoRa, to transmit an alarm signal along with coordinates of the event to the central location. Although this is conceptually possible, this option was left outside the scope of the paper since it represents a very specific scenario (that the authors unfortunately did not considered). Nonetheless, taking the size of the LoRa sensor into account, the main concept discussed in the paper will not be affected: instead of using a camera as a sensor to gather “nectar” information, a LoRa sensor can be carried, and activated when detection is made. We also included a corresponding explanation in the introduction of the new version of the manuscript.
*Specify that you might need to create a new prototype with a cheaper drone to truly test it to be cost effective. Right know you worked with ideal conditions with a top of the line multirotor. Take care of those Hummingbirds.
We fully agree with the reviewer in this aspect. As we previously justified, our decision to use Hummingbirds for the experiments was essentially motivated by the drones we currently have at our disposal. Again we would like to remark that we could use a custom-built drone or any other drone in the market that fulfills the basic requirements pointed out in Sec. 4. These basic requirements definitely permit a cost-effective solution. We included this clarification in the new version of the manuscript.
*Your POI and ROI path planning still not guarantee immediate safer path but is a good work in progress. I am glad you were honest about it. That approach can be a great application into and Unmanned Traffic System that can be implemented for a dynamic airspace or airway cell risk evaluator.
We fully agree with the reviewer. Our main intention in this work was also to highlight the current limitation of our system. This will allow us and other researchers to further develop the proposed system.
Reviewer 3 Report
This is a systems paper presenting an overview of a complete, integrated multi-UAV HW/SW platform for autonomous area inspection, meant to aid first responders in emergency situations. It consists of a central on-ground coordination stations ("beehive") and a swarm of UAVs ("bees") that collect data ("nectar"). The underlying assumptions and problem formulation (static pre-mapped environment, no autonomous target tracking) limit system capabilities on one hand, but allow the authors to come up with a cheap and efficient solution that doesn't involve inter-UAV transactions via expensive long-range communication links, or highly accurate differential GPSs.
There is not much to comment about the paper. It is well-written, well-organized and relatively complete for what it is: a comprehensive, high-level final project report. Despite the limited scientific value, it should be of technical interest to UAV platform designers. The choices made seem to be in-line with the current state-of-the-art, given the problem formulation and the stated goals.
From a robotics perspective, the most relevant part would be the graph-based algorithm enabling the Map Discretization module, which seems to be somewhat novel. My guess though would be that it has already been published on its own, but the reference is missing.
The authors state that the logistics of managing large UAV swarms are currently prohibitive, thus the system is practically limited to 10 drones. What are the specific SW/HW obstacles that hinder system scaling to hundreds of UAVs? Please elaborate.
Also, why the scalability analysis in the experimental section is constrained only to 4 drones? This is too limited, since the authors state that up to 10 UAVs are easily supported. Evaluating system performance up until (at least) 7 drones would be more convincing. Of course, it is mentioned that "having a number of drones larger than the number of regions does not bring a higher performance to the system", but it is not enough. If it is true, more regions should be used for evaluating larger swarms. This would also strengthen the authors' claim that the system is robust to occasional drone failures, since more comprehensive experiments could be performed towards this direction.
Author Response
This is a systems paper presenting an overview of a complete, integrated multi-UAV HW/SW platform for autonomous area inspection, meant to aid first responders in emergency situations. It consists of a central on-ground coordination stations ("beehive") and a swarm of UAVs ("bees") that collect data ("nectar"). The underlying assumptions and problem formulation (static pre-mapped environment, no autonomous target tracking) limit system capabilities on one hand, but allow the authors to come up with a cheap and efficient solution that doesn't involve inter-UAV transactions via expensive long-range communication links, or highly accurate differential GPSs.
We would like to thank the reviewer for the time devoted to our work. We highly appreciate the constructive comments. Next we address each of the critic points raised by the reviewer.
There is not much to comment about the paper. It is well-written, well-organized and relatively complete for what it is: a comprehensive, high-level final project report. Despite the limited scientific value, it should be of technical interest to UAV platform designers. The choices made seem to be in-line with the current state-of-the-art, given the problem formulation and the stated goals.
Thank you very much for the positive and encouraging comment.
From a robotics perspective, the most relevant part would be the graph-based algorithm enabling the Map Discretization module, which seems to be somewhat novel. My guess though would be that it has already been published on its own, but the reference is missing.
Thanks for the valuable comment. In fact, the map discretization module is novel, and has not been published before. We think that this article is the best venue to publish our method. Within the context of the overall system it is much easier to motivate the chosen approach compared to a publication that just focuses on the Map Discretization.
We added some text in the paper’s conclusion to highlight that we propose a new method for the map discretization.
The authors state that the logistics of managing large UAV swarms are currently prohibitive, thus the system is practically limited to 10 drones. What are the specific SW/HW obstacles that hinder system scaling to hundreds of UAVs? Please elaborate.
Thanks for the comment. This is indeed a point that should be further elaborated. In general there a few constraints that make it difficult to scale the system up.
Communication: The communication only takes place around the hive. There the UAVs upload their collected information, e.g. images to the database. However, the bandwidth and data rate of the communication is limited, so that it would not be possible for a couple of dozen UAVs to upload their data at the same time. Landing Area: In our system each drone has a dedicated landing spot. In the worst case all UAVs can arrive at the same time at the hive and land at the same time. For hundreds of UAVs a large area would be needed just for starting and landing, since each UAV needs its own spot. Maintenance: The currents system with up to 10 UAVs has the nice feature that one operator can handle it. If UAVs arrive back at the base station the operator has to change the batteries of the UAV. For hundreds of UAV you would need again multiple operators for that job, because many UAVs may arrive at the same time.We added a paragraph to section 3.3 to elaborate on this issue.
Also, why the scalability analysis in the experimental section is constrained only to 4 drones? This is too limited, since the authors state that up to 10 UAVs are easily supported. Evaluating system performance up until (at least) 7 drones would be more convincing. Of course, it is mentioned that "having a number of drones larger than the number of regions does not bring a higher performance to the system", but it is not enough. If it is true, more regions should be used for evaluating larger swarms. This would also strengthen the authors' claim that the system is robust to occasional drone failures, since more comprehensive experiments could be performed towards this direction.
We fully agree with the reviewer that having experiments with more drones can provide interesting data. However, this is really challenging. First, because we do not have a fleet of 7 drones at our disposal. In addition, it is very hard to find an area that is large enough to have 7 drones flying at the same time while covering a reasonable area. In our case, the obstacles are limiting this factor. Besides, due to regulations in Germany, it is necessary to have a safety pilot for each drone. That implies that for each experiment you need at least 7 people holding the remotes of each drone, 1 person coordinating and 1 person taking care of the base station. This requires extra resources as transportation, working hours and equipment.
As a viable solution, we decided to analyze in simulations the system performance with a larger number of UAVs. In particular, we considered up to 8 UAVs. We added a section in the revised manuscript (Sec. 5) with this evaluation. We are now convinced that the balance between simulation and experimental results permits us to draw definitive conclusions about the proposed system design.
Round 2
Reviewer 1 Report
I believe that authors have clarified all the possible problems in the previous version. Finally, I consider that the m/s is ready for publication. Congratulations to the authors.
Reviewer 3 Report
The authors have appropriately responded to the review comments. I recommend acceptance.